# Genome-Wide Sequencing Modalities for Children with Unexplained Global Developmental Delay and Intellectual Disabilities—A Narrative Review

**DOI:** 10.3390/children10030501

**Published:** 2023-03-03

**Authors:** Mary Hsin-Ju Ko, Hui-Ju Chen

**Affiliations:** 1Division of Pediatric Neurology, Department of Pediatrics, Hsinchu Municipal Mackay Children Hospital, Hsinchu 300, Taiwan; 2Division of Pediatric Neurology, Department of Pediatrics, MacKay Children Hospital, Taipei 252, Taiwan; 3Department of Medicine, Mackay Medical College, New Taipei 252, Taiwan

**Keywords:** next-generation sequencing, intellectual disability, developmental delay, exome sequencing, genome sequencing, chromosomal microarray, neurological developmental disorder, global developmental delay

## Abstract

Unexplained global developmental delay (GDD) and intellectual disabilities (ID) together affect nearly 2% of the pediatric population. Establishing an etiologic diagnosis is crucial for disease management, prognostic evaluation, and provision of physical and psychological support for both the patient and the family. Advancements in genome sequencing have allowed rapid accumulation of gene–disorder associations and have accelerated the search for an etiologic diagnosis for unexplained GDD/ID. We reviewed recent studies that utilized genome-wide analysis technologies, and we discussed their diagnostic yield, strengths, and limitations. Overall, exome sequencing (ES) and genome sequencing (GS) outperformed chromosomal microarrays and targeted panel sequencing. GS provides coverage for both ES and chromosomal microarray regions, providing the maximal diagnostic potential, and the cost of ES and reanalysis of ES-negative results is currently still lower than that of GS alone. Therefore, singleton or trio ES is the more cost-effective option for the initial investigation of individuals with GDD/ID in clinical practice compared to a staged approach or GS alone. Based on these updated evidence, we proposed an evaluation algorithm with ES as the first-tier evaluation for unexplained GDD/ID.

## 1. Introduction

Childhood neurodevelopment is complex and is interwoven with genetic, biologic, and environmental factors. A simplified schema of developmental domains consists of cognitive, language, socio-emotional, and motor neurodevelopment. Global developmental delay (GDD) is defined by significant developmental delay in at least two developmental domains, with performance more than two standard deviations below the mean. Individuals with intellectual disability (ID) usually have deficits in adaptive behaviors and cognitive abilities present from an early age. ID is a clinically defined condition with an intelligent quotient (IQ) under 70 on the Wechsler Intelligence Scale for Children (WISC) or DQ < 76 in two or more developmental domains assessed by the Gesell Developmental Scale. GDD pertains to children under five years of age, while ID pertains to children aged five years or older. GDD/ID can be secondary to known genetic or chromosomal disorders or be secondary to known antenatal or perinatal insults of the brain. However, the majority of GDD/ID remains idiopathic, or unexplained. Together, unexplained GDD/ID have an estimated prevalence of 1 to 2%.

While functional diagnoses of GDD/ID are established by clinical evaluations and standardized assessments, etiologic diagnoses of GDD/ID often require intricate investigations. Clinically, functional diagnoses and serial evaluations provide insights regarding the efficacy of therapeutic strategies. Etiologic diagnoses, on the other hand, determine the therapeutic goal and provide prognostic information. Having an etiologic diagnosis also brings an end to the diagnostic odyssey, which is a process that often results in a sense of uncertainty and anxiety for both the patients and their family [1]. Despite the established diagnostic evaluation protocols and recommendations of GDD/ID, nearly half of the individuals with GDD/ID do not have an etiological diagnosis. Although inborn errors of metabolism (IEM) are rare, IEM investigations used to be the first steps in the evaluations of GDD/ID etiologies. However, the diagnostic rate of IEM workups in individuals with GDD/ID was under 1%. It is estimated that nearly half of unexplained GDD/ID cases have genetic causes [2]. Therefore, the deployment of diagnostic genetic testing will facilitate the search for etiologic diagnosis of unexplained GDD/ID and improve the diagnostic rates significantly.

The clinical utility of genetic testing has been bolstered over the past two decades, making genetic testing an indispensable asset in the evaluation of children with rare conditions. Unexplained GDD/ID lack characteristic clinical features, and reaching an etiologic diagnosis often depends on genotype-driven analysis. Conventional G-banded karyotyping identifies large segments (≥5 MB) of chromosomal aberrations, in addition to aneuploidy. With the advent of cytogenetic technologies, single-gene disorders and small segment genetic perturbations can be detected by massive parallel sequencing (MPS)-based molecular modalities including targeted gene panels (TGP), exome sequencing (ES), and genome sequencing (GS).

Following the American Council of Medical Genetics and Genomics (ACMG)’s recommendation in 2010, chromosomal microarrays (CMA) have become the first tier of genetic testing for individuals with unexplained GDD/ID, autistic spectrum disorder, and/or multiple congenital anomalies (MCA) [3,4,5]. Recently, high-throughput sequencing, such as ES or GS, has allowed for rapid, in-depth analysis of the human genome. New evidence is showing support for MPS to replace CMA as the first-tier test for unexplained autistic spectrum disorder, GDD/ID, and MCA [6,7,8,9]. ACMG has also published an updated guideline to recommend ES and GS as the first-tier genetic evaluation for patients with ID [10].

The objective of this review is to examine current evidence on the clinical utility of MPS in individuals with unexplained GDD/ID.

## 2. Genetic Etiologies of Intellectual Disabilities and Global Developmental Delay

Currently, the candidate and causative GDD/ID genes are rapidly accumulating, and more than 2500 ID genes have been identified [11]. The Deciphering Developmental Disorder project, which is a large, population-based study on unexplained GDD/ID in the United Kingdom and Ireland, has identified 14 new genes associated with GDD/ID [12]. The top 10 most prevalent DNV among GDD/ID are listed in Table 1 [2]. Most of the GDD/ID genes discovered were de novo variants (DNV) [2,8,13,14]. Deletions were more common than duplications, resulting in frameshift, stop-gained, alternative-splice-site, or missense mutations. Autosomal recessive ID genes were mostly discovered in individuals born to consanguineous family. These genes may also be seen as DNV in compound heterozygosity. In addition to nuclear variants, putative mutations in mitochondrial DNA have also been implicated in ID [15]. Although ID is more prevalent in males, contributions of X-linked genes are similar in both sexes (~6%) [16]. Some of the X-linked ID genes exhibit skewed inactivation and incomplete penetration, resulting in varying degrees of clinical phenotypes in females [17]. ID genes with female-biased burdens and extreme skewing that have been identified to date include *DDX3X*, *MECP2*, *WDR45*, *SMC1A*, *HDAC8*, and *NHS* [13,17].

The GDD/ID variants are classified based on the ACMG five-tier criteria: pathogenic, likely pathogenic (LP), variants of unknown significance (VUS), likely benign, and benign variants [32]. Currently, the terms are applicable for the results of CMA, ES, and GS. Pathogenic and LP variants are considered causative by most studies. The pathogenic or LP genetic variants of GDD/ID include large segment aberrations, such as autosomal and sex chromosomal aberrations, and small genic variations, such as monogenic mutations, microdeletions and/or microduplications, balanced structural variants, trinucleotide repeat expansions or short tandem repeats, uniparental disomy, and mobile element insertion. Table 2 lists some of the key terminology frequently encountered in the results of genetic testing reports.

These genetic variations may be present in both coding and non-coding DNA regions and have different degrees of penetrance and inheritance. The current variant calling and filtering algorithms for genome-wide testing were designed to identify causative genes with monogenic or Mendelian inheritance [33,34]. Genes with complex or incomplete inheritance would require further interpretation if the results were not compatible with the proband phenotypes and clinical suspicion. More than often, a clinical phenotype may be associated with multiple candidate variants, and the genotype association would mandate validation or segregation analysis by testing both the affected child and the unaffected parents. Trio sequencing or traditional molecular methods such as single-gene testing or direct Sanger sequencing may help delineate the causative genes.

Overall, the diagnostic yield of molecular modalities is positively correlated with the number of dysmorphic features or MCA that are present [8]. The more profoundly the IQ of an individual is affected, the more likely the ID is to have a genetic etiology. Earlier onsets of GDD were also more likely to have genetic defect associations. Many studies have attempted to establish the genotype and phenotype correlations in individuals with unexplained GDD/ID; however, the heterogenicity of pathogenic variations and clinical presentations made the association challenging. Individuals harboring GDD/ID variants may present with phenotypes unrelated to GDD/ID [35]. Early detection via phenotyping could also be challenging, as some of the GDD/ID age-dependent symptoms evolve over time. The Human Phenotype Ontology (HPO) and gene-to-phenotype (G2P) programs are currently under development to build phenotype models of genetic variants. These models will facilitate in silico phenotype–genotype matching and help augment the current variant calling and interpretation algorithms.

## 3. Genetic Diagnostic Tools for Unexplained Intellectual Disabilities and Global Developmental Delay

### 3.1. Chromosome Microarray

There are two different CMA platforms: array comparative genomic hybridization (aCGH) and single-nucleotide polymorphism arrays (SNP). aCGH compares an individual’s DNA with that of a control sample and labels the differences between the two sets of DNA. SNP compares the different alleles of the patient’s own DNA. It is especially useful in identifying uniparental disomy, consanguinity, or ancestral homozygosity. Both CMA platforms allow for identification of variations beyond disorders of chromosomal numbers and ploidy. They are also able to detect small genetic variations at least 1kB in size resulting from gains (microduplications) or losses (microdeletions) of the genomic DNA.

The diagnostic yield of CMA varied from 11 to 36% (Table 3). The diagnostic yield was associated with not only diagnostic modality but also the study design, especially the cohort phenotypes. Overall, the diagnostic rate was positively correlated with the severity of ID and number of dysmorphic features present. CMA is thus an ideal genetic testing option for individuals presenting with unexplained GDD/ID and MCA.

The strength of CMA lies in its ability to detect small chromosomal imbalances beyond the scope of karyotyping. Furthermore, the automated analysis of the microchips and also the shortened turnaround allow for rapid and timely analysis. However, CMA has limited ability to detect balanced chromosomal rearrangements, trinucleotide repeat expansions, single-nucleotide variations, and low-level mosaicism finer than its resolution. Although the overall diagnostic rate of CMA is superior to that of conventional karyotyping, CMA provides etiologic diagnoses to less than one-third of the individuals with unexplained GDD/ID. CMA also does not analyze the mitochondrial genome. Therefore, in order to overcome the hurdle of limited diagnostic yield, advanced techniques addressing these shortcomings of CMA are needed.

### 3.2. Massive Parallel Sequencing

MPS, commonly referred to as next-generation sequencing (NGS), allows for genome-wide analysis. It has become the mainstay of genetic testing in recent times and has enabled rapid progression in the recognition of gene–disorder associations.

#### 3.2.1. Gene Panels

Gene panels utilize NGS technology to decode the genome and to analyze and interpretate the set of predetermined “core genes”. The candidate genes in the panels are tailored to facilitate the detection of a specific set of disorders, syndromes, or phenotypes. They usually include the known disease-causing genes, though VUS are sometimes selected too. There are two types of gene panels: targeted gene panels (TGP) and virtual panels. TGP, sometimes referred to as “partial exome testing”, focus on the candidate genes and do not test the remainder of the genome. The candidate genes are enriched through hybridization probe capture or PCR amplification, which increases the depth of sequencing. The probe numbers and sites can also be customized to include variants under suboptimal coverage in ES and GS. TGP has been shown to be superior to ES and GS for neuromuscular disorder and congenital retinal diseases [54]. On the contrary, virtual gene panels utilize ES or GS libraries and interrogate the genes of interests in silico. Future iterative analysis is thus possible with virtual gene panels.

Table 4 lists recent literature on GDD/ID TGP testing. The diagnostic yield ranged from 8% to 63%. Although the diagnostic yield of panel testing was previously shown to be independent from the numbers of candidate genes tested, the discrepancies in diagnostic rates between studies could be secondary to study design, testing, and interpretation algorithms [55]. Angione et al. proposed a four-point scoring system based on the core determinants of panel efficacy [56]. The evaluating scheme included minimal and average depth of coverage, breadth of deletion/duplication, cost, turnaround time, and the availability of free parental testing for VUS resolution. The lower the total scores, the better the panel design. Application of the scoring scheme would allow ordering providers to compare and rank diverse panel designs.

Because TGP data are generated on a finite number of genes, the targeted analysis approach focuses on clinically relevant information, minimizing secondary findings. In turn, it allows for rapid data analysis at a much lower cost than ES or GS. Focused investigations also result in higher sensitivity, with better detection of mosaicism. However, there are currently no clinical guidelines for or definitions of an “ideal” panel design. The candidate genes selection is usually customized and very heterogeneous among different laboratories. Different platforms also utilize different molecular techniques, with varying degrees of depth of coverage. Reanalysis of data with an updated genomic library is not possible after TGP testing, because it is fixed in the number of genes originally accessed.

#### 3.2.2. Exome Sequencing

Exomes are the protein-coding regions of the genome. They make up only about 1–2% of the whole genome but encompass nearly 85% of the disease-related genes. The widespread use of ES recently has allowed rapid expansion of the sequencing library and refinement of existing analysis protocols. The analytical steps, including the primary filtering variant calling pipeline and the secondary filtering in silico predictive programs, have been augmented to interpretate the coding regions and their vicinity efficiently. Because GS is just starting to gain popularity, the bioinformatics and the associated data managements of GS are relatively immature compared to those of ES [33].

A consensus statement by Srivastava et al. and the NDD Exome Scoping Review Work Group has suggested ES be the first-tier clinical diagnostic test for individuals with neurodevelopmental disorders [61].

The diagnostic yield of ES in GDD/ID individuals with normal genetic screening results ranged from 21 to 66% (Table 5). A recent meta-analysis revealed the average diagnostic yield of ES in individuals with unexplained ID to be 42% [62]. The diagnostic yield was higher for individuals with trio or familial ES results [63,64]. Studies on individuals from consanguineous parents also had a higher yield. Many laboratories are now including CNV calling algorithms in ES analysis, which can further improve the resolution of ES [65]. Most of the literature reviewed had enrolled probands with prior negative molecular screening, including CMA. The yield would be even higher if ES was the first-tier investigation. At the rapid accumulation of GDD/ID genes, ES is expected to provide answers to at least 50% of unexplained GDD/ID as the reference database continues to expand [66].

Unlike CMA, which detects regional or segmental perturbations, ES detects single-gene variations and provides a genetic diagnosis. Establishment of a definitive diagnosis can direct disease therapy, give prognostic information, and contribute to fertility planning. One meta-analysis study has demonstrated the cost-effectiveness of ES [10]. It is able to provide higher diagnostic yield at lower costs when used in the initial investigation for GDD/ID etiology instead of being used after staged and extensive testing. Although sequencing is no longer the rate-limiting step in ES, the diagnostic efficacy still depends on the platform design, which includes the number of probes utilized to capture the region of interest within the coding region. ES is unable to cover all the exomes and is prone to having false negative gaps in GC-rich regions, repetitive regions, regions with pseudogenes, and regions with high homology [80]. Additionally, low probe affinity for the highly variable regions and missed coverage could both lead to false negative results [7]. The regions with reduced coverage depend on sequencing metrics and may differ between labs.

#### 3.2.3. Genome Sequencing

GS provides coverage of both CMA and ES and uncovers variants missed by CMA and ES including structural variations, trinucleotide repeats, and mitochondrial variants in one experiment. Medical GS is only starting to gain popularity in recent years, and expert consensus and guidelines are beginning to be formulated and published [33,34,81,82]. Currently, GS is mostly based upon short-read technology, meaning the genome must be fragmented and cloned into overlapping segments with 75 to 145 base pairs. The segments are then aligned and sequenced by computer programs to reproduce the genome. However, the highly repetitive regions of the human genome can be challenging for short-read sequencing to decipher [83]. Some labs are beginning to adapt the “third” generation of technology with longer reads, which can read between 5 kilobase pairs and 30 kilobase pairs [83,84]. Currently, the long-read sequencing technology has a high error rate, and its clinical efficacy in diagnosing GDD/ID is still yet to be assessed.

Studies on using GS to evaluate unexplained GDD/ID are limited. Recent publications utilizing GS on individuals with GDD/ID have demonstrated the diagnostic yield ranged between 21 to 63% (Table 6). Overall, compared to ES, the extrapolation of the additional information by GS is estimated to enhance the diagnostic rate of ES by 9% to 15% [85,86]. As GS becomes more clinically available, it is expected to gradually replace ES and CMA. In some countries and limited laboratories, GS has become the first-tier genetic evaluation for individuals with GDD/ID. The French Genomic Medicine plan is conducting a DEFIDIAGE pilot study that implements GS of individuals with unexplained GDD/ID [87]. It is a nation-wide study and will be carried out in 15 centers across France. The diagnostic yield of GS is expected to improve as GS technology continues to evolve.

GS has the potential to become the first-tier etiologic evaluation of GDD/ID, especially in identifying short segment variants missed by CMA, and CNV and non-coding segment deletions or allele dropouts missed by ES [7]. The challenge in GS is the variant prioritization and interpretation of these complex, rare, and non-coding region variants. The GS platform designs utilized in the current literature are still very diverse, further complicating report interpretation. Relative to ES, GS technology is still expensive, and the workflow is lengthy and time-consuming, yset the efforts only result in a modest increase in diagnostic yield, limiting its clinical feasibility. While GS has been shown to detect an additional 34% of disease-causing variants in ES-negative individuals, ES reanalysis would have detected 30–50% of these GS-positive variations, resulting in an overall 7 to 9% difference in the diagnostic yield of GS and ES [85,90].

#### 3.2.4. Beyond Genomic Sequencing

The “omics” are alternatives to GS or ES in identifying the genetic etiology of rare diseases [91,92]. The “omics” include transcriptomics, which involve RNA sequencing; metabolomics, which uncover disease mechanisms; and proteomics, which reveal impairments in protein synthesis, stability, degradation, and signaling. These techniques also provide functional evidence to verify the pathogenicity of the variants identified by whole-genome analysis. In addition, methylation profiling may elucidate epigenetics’ effects on the DNA. The clinical utility of these molecular technologies in the diagnosis of unexplained GDD/ID is yet to be studied.

## 4. Discussion

This review described the current status of genetic testing for GDD/ID and the key components to consider when deciphering the genome, including the diagnostic yield, strengths, and limitations of commonly used methods. The diagnostic yields of CMA, TGP, ES, and GS for GDD/ID were 11–36%, 8–63%, 21–66%, and 21–63%, respectively. To summarize, ES and GS outperformed CMA and TGP in identifying GDD/ID-related variants. The large discrepancies in diagnostic yield could potentially be attributed to the heterogenous study design and participant inclusion criteria. Table 3, Table 4, Table 5, Table 6 list the inclusion criteria of studies reviewed. The clinical phenotypes of the participants varied. For example, some studies recruited subjects with any NDD versus subjects with GDD, and some included participants with or without ASD. The participants also had different prior genetic evaluation results. Some studies only included subjects with negative genetic testing, while some tested all participants irrespective of previous test results. Meta-analysis should be performed to provide an objective comparison of diagnostic performance.

With the advent of cytogenetic analysis, genome-wide sequencing is becoming an integral part of medical practice. The diagnostic yield and the cost-effectiveness of the analytical methods are perpetually being refined and enhanced. Studies on health economics have shown the costs and benefits of ES and GS. Prior to receiving an etiologic diagnosis by ES, a family with an ID individual would have spent on average $16,409 for the total diagnostic process. The cost of trio ES was 42% of the spending [93]. In comparison to singleton ES, trio ES had better, though statistically insignificant, diagnostic yield [64]. Analysis of trio ES also resulted in fewer variants needed for curation and co-segregation evaluations and less VUS in ES-negative individuals. However, trio ES is double the cost of singleton ES [64]. Given that diagnostic yield depends on the phenotypes, if sufficient information is provided for the variant prioritization, singleton ES would be the most cost-effective option. On the contrary, in individuals with complex phenotypes and a low likelihood of monogenic disorders, trio ES would be a better option in terms of diagnostic cost and efficacy [64]. Ewans et al. suggested ES offers the lowest-cost pathway for individuals with or without prior ES testing, while GS possesses the maximal diagnostic potential [90]. The costs of MPS are expected to decrease in the near future as the associated molecular technology matures. The financial feasibility also varies in different countries and clinical settings. It is therefore a shared decision made by the ordering clinician and the patient/family, after evaluating all trade-offs and available resources. Genetic tests are being ordered by not only geneticists but also pediatric neurologists, developmental specialists, and even primary care physicians. Although genetic counselling is indispensable, ordering providers should be able to interpretate the results of the genetic tests. Understanding the strength and limitations of different modalities is thus important for pediatricians before ordering the tests.

With the high-throughput analysis of large DNA segments, VUS are often detected. Additionally, LP variants can still have a 5% to 10% chance of being false positives. The diagnostic steps therefore do not stop at laboratory classification. Clinical judgment should always follow. VUS, after careful triage and reevaluations, could be reclassified as pathogenic or LP variations. Periodic iterative analysis is also important: reanalysis of the ES data can increase the diagnostic yield from 11% to 15% [66,71,94,95,96]. It is recommended that inconclusive results be reevaluated every 6 to 12 months [61,95,96]. However, it is the ordering provider’s responsibility to retrieve the affected cases and to request the associated laboratories to reanalyze the data [81]. The testing laboratory is not expected to offer proactive updates.

SEQC2 is a genome-sequencing quality control project led by the FDA in the United States to evaluate the inter-platform reproducibility of NGS technology and to establish best-practice recommendation [97]. Nevertheless, the sequencing procedures and analysis algorithms for GS and ES have not been well standardized. Different laboratories have different exome capture kits, sequencing platforms, variant calling pipelines, and realignment and in silico predictive programs. These differences may result in inconsistent depth of coverage and interfere with the final interpretation. Pathogenic variants usually have low allele frequencies, and detection probability relies on the depth of coverage, which is defined by the number of reads. There is no consensus on the number of mutated reads in MPS thus far. Specific probe targeting strategies need to be deployed in order to minimize inconclusive results. Standardized test platforms, analytical algorithms, and quality assessment for GS are also still in progress. Ordering clinicians must be aware of these limitations when interpretating the reports.

The current guidelines on MPS have not emphasized the orthogonal validation procedures of the reportable variants. The diagnostic accuracy should be the intrinsic property of the sequencing platform and bioinformatic metrics. Furthermore, the results of MPS have been shown to be concordant with those of Sanger sequencing, with only 0.3% discrepancy [98]. Consequently, implementation of Sanger sequencing to validate the MPS results is not recommended [81,98]. However, the accuracy of MPS in detecting complex variants, such as trinucleotide repeat expansion and structural variants, has not been verified, and confirmation is necessary. Service providers should therefore be able to determine the variant types needing confirmatory testing before reporting [33].

Apart from wet and dry laboratory techniques, ethical issues should also be considered in the search for a genetic etiology. Genome-wide analysis inadvertently and inevitably brings about secondary findings. These include disease-causing genes irrelevant to the patient’s phenotypes and other undesirable harms associated with the disclosure. Equity in resource allocation should also be discerned. Although the benefit of establishing a genetic diagnosis is apparent, resources are limited for marginalized groups and in developing countries [10]. Many developing countries are only starting to adopt CMA as the first-tier evaluation for GDD/ID [36,45]. It is, again, a shared decision-making process for the patient/family and the ordering provider. Informed consent and options for variant interpretation and secondary findings should be available for all parties.

### Proposed Evaluations Algorithm for Unexplained Intellectual Disabilities or Global Developmental Delay

Based on the literature reviewed, we agreed with the ACMG recommendations that genome-wide analysis should be the first-tier evaluation for individuals with GDD/ID. Balancing the effects of diagnostic ability and financial feasibility, we believe ES is the more cost-effective testing option to be deployed in a medical setting. Among the different NGS-based genetic analyses, we strongly recommend ES to be the standard, first-tier evaluation for the etiologic diagnosis of GDD/ID (Figure 1).

## 5. Conclusions

The current literature and guidelines support the use of ES as the first-tier investigation in individuals with unexplained GDD/ID. Inconclusive studies may be followed by GS, karyotyping, or iterative analysis.

## Figures and Tables

**Figure 1 children-10-00501-f001:**
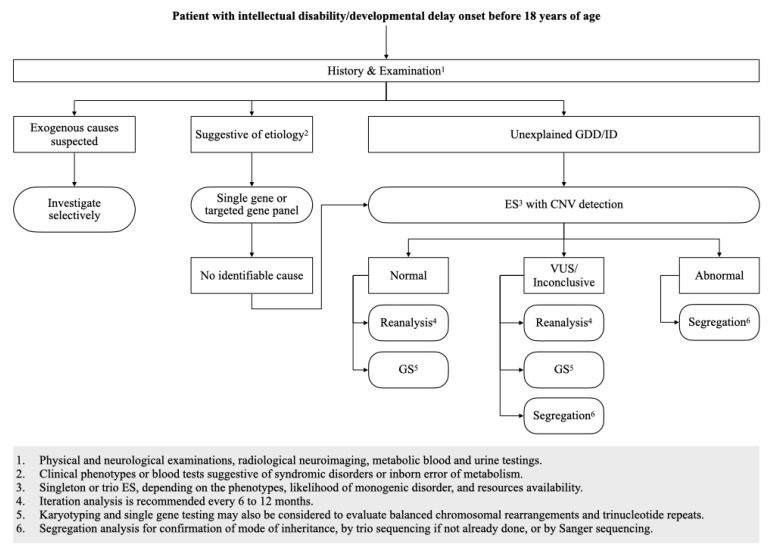
Proposed diagnostic algorithm for evaluation of unexplained GDD/ID. Abbreviations: CNV = copy number variants; DD = developmental delay; ES = exome sequencing; GS = genome sequencing; ID = intellectual disability; VUS = variants of unknown significance.

**Table 1 children-10-00501-t001:** Common de novo disturbing variations associated with global developmental delay/intellectual disability.

Gene	Location	Phenotypes Other than GDD/ID	Ref.
*ARID1B*	6q25.3	Coffin–Siris syndrome	[2,18]
*DDX3X*	X	Verbal dyspraxia, hypotonia Mostly in females; rare in males	[2,19]
*KMT2A*	11q23	Wiedemann–Steiner Syndrome, characteristic dysmorphism	[2,20]
*DYRK1A*	21q22.13	Characteristic facial features, feeding difficulty, speech impairment, microcephaly, epilepsy	[2,21]
*CTNNB1*	3p22.1	Exudative vitreoretinopathy, truncal hypotonia, peripheral spasticity, microcephaly	[2,22]
*ADNP*	20q13.13	Syndromic autism, dysmorphic facial features, seizure, hypotonia, early tooth eruption	[2,23]
*STXBP1*	9q34.11	Early infantile epileptic encephalopathy 4 (EIEE4), epilepsy, behavior problems, movement disorders	[2,24,25,26]
*SCN2A*	2q24.3	Epilepsy syndromes, non-syndromic ID, ASD	[2,27,28]
*MED13L*	12q24.2	Distinctive facial features with macroglossia, macrostomia, congenital heart defects	[2,29,30]
*SATB2*	2q33.1	SAS syndrome, hypotonia, feeding difficulty, craniofacial anomalies	[2,31]

**Table 2 children-10-00501-t002:** Definitions of Key Terminology in Genetic Molecular Analysis.

Terms	Definition
Recurrent copy number variant (CNV)	Genetic rearrangements that recur in multiple individuals, with similar length and breakpoint.
Non-recurrent CNV	Genetic rearrangements with scattered breakpoints and different sizes that are usually different among different individuals.
Single-nucleotide variants (SNV)	Substitution of a single nucleotide for another. The exchange is non-synchronous if the SNV results in a change in amino acid, and synchronous if the SNV does not result in a change in amino acid. The SNV can also be a stop gain, resulting in premature termination of protein transcription.
Small insertions and deletions (Indels)	Insertion or deletion of less than 50 base pairs length of DNA, often resulting in frameshift changes.
Structural variants	Changes in the DNA length of greater than 50 base pairs, including deletion, duplication, inversion and translocation. Copy number variants are examples of imbalanced structural variants.
Runs of homozygosity (ROH)	Continuous homozygous DNA segments in diploid genomes, commonly used to diagnose uniparental isodisomy, consanguinity, and replicative DNA repair events.
Repeat expansions/short tandem repeats (STR)	Trinucleotide repeat expansions that are unstable mutations and increase in size in the successive generations.
Mitochondrial variants	Changes similar to nuclear genomic variations, including SNV, indels, and structural variants. Additionally, if heteroplasmic, analytical validity must be carefully reviewed with clinical phenotypes.
Mosaic variants	Genetic variations that occur after fertilization, resulting in two or more genetically different cell lines.
Null variants	Canonical nonsense or frameshift deletion, resulting in loss of function in a gene
Variant calling	The process of variant identification, which is an integral part of genetic assessment
In silico predictive programs	Computational analysis tools that aim to prioritize variant triage and to determine the potential effect of the sequence variant on the gene transcript and the protein products.
Sequencing coverage	Number of reads that covers a DNA segment and is defined by the Lander/Waterman equation: C = LN/G (C is coverage, L is the read length, N is the number of reads, and G is the haploid genome length).
Variant allele frequency (VAF)	Prevalence of a specific gene within a population. Variants associated with rare conditions typically have low allele frequencies (<1%). To increase the sensitivity of the variant calling algorithm, the threshold of allele frequency is usually set higher, from 3% to 10%.
Diagnostic yield	Proportion of individuals carrying a pathogenic/likely pathogenic variant in a cohort.

**Table 3 children-10-00501-t003:** Chromosomal microarray analysis of individuals with intellectual disabilities/developmental disabilities in recent years.

Study	Country	N	Cohort Phenotype	DY	Ref.
Subjects with Normal Karyotypes					
Leite et al. (2022)	Brazil	83	GDD/ID +/− MCA	33%	[36]
Levchenko et al. (2022)	Russia	91	Non-specific ID	18%	[8]
Liu et al. (2022)	China	251	Unexplained ID/DD	32%	[37]
Yuan et al. (2021)	China	2688	Non-syndromic ID/DD	21%	[38]
Espeché et al. (2020)	Argentina	133	ID with dysmorphic features	12%	[39]
Lee et al. (2019)	Taiwan	177	ID/DD	32%	[40]
Wang et al. (2019)	China	358	Isolated ID/DD	25%	[41]
Chan et al. (2018)	Hong Kong	138	Moderate–profound GDD/ID	12%	[42]
Chen et al. (2018)	China	60	Moderate–severe ID	20%	[43]
Kim et al. (2018)	Korea	50	Severe ID/DD	36%	[44]
Subjects Without Karyotype Results					
Kamath et al. (2022)	India	67	GDD/ID +/− comorbidities	21%	[45]
Miclea et al. (2022)	Romania	189	GDD/ID +/− comorbidities	19%	[46]
Chen et al. (2021)	Taiwan	61	Unexplained moderate–severe ID/DD	20%	[47]
Ogûz et al. (2021)	Turkey	302	GDD/ID +/− abnormal head size, behavior, epilepsy activity	11%	[48]
Yang et al. (2021)	Korea	308	Unexplained ID/DD +/− MCA	19%	[49]
Arican et al. (2019)	Turkey	210	Unexplained ID/DD	12%	[50]
Hu et al. (2019)	China	332	Isolated ID/DD	18%	[51]
Ceylan et al. (2018)	Turkey	124	GDD/ID	17%	[52]
Pinheiro et al. (2020)	Spain	215	Unexplained GDD/ID	23%	[53]

Abbreviations: DD = developmental delay; DY = diagnostic yield; GDD = global developmental delay; ID = intellectual disability; MCA = multiple congenital anomalies; N = number of cases with intellectual disability/global development delay/developmental delay. Ref. = Reference.

**Table 4 children-10-00501-t004:** Targeted gene panels of individuals with intellectual disabilities/developmental disabilities in recent years.

Study	Country	N	Previous Investigation	Cohort Phenotype	No. Genes	DY	Ref.
Leite et al. (2022)	Brazil	19	CMA, K	GDD/ID +/− MCA	1252	63%	[36]
Ibarluzea et al. (2020)	Spain	47	K, Fragile X	Male with unexplained GDD/ID	82 *	26%	[57]
Pekeles et al. (2019)	Turkey	48	CMA, K	GDD/ID	143−2308 **	21%	[58]
Yan et al. (2019)	China	112	Nil	Unexplained ID/DD	454	8%	[59]
Gieldon et al. (2018)	Germany	106	CMA, K	Unexplained ID/DD +/− MCA	66	34%	[60]
Han et al. (2018)	Korea	35	CMA	Unexplained ID/DD	4813	29%	[9]

* X-linked genes only; ** Four different panels. Abbreviations: CMA = chromosome microarray; DD = developmental delay; DY = diagnostic yield; GDD = global developmental delay; ID = intellectual disability; K = karyotyping; N = number of cases with intellectual disability/global development delay/developmental delay; No. Genes = number of candidate genes included in the panel; Ref. = reference.

**Table 5 children-10-00501-t005:** Exome sequencing of individuals with intellectual disabilities/developmental disabilities in recent years.

Study	Country	N	Previous Investigation	Cohort Phenotype	DY	Ref
Studies with singleton approach						
Al-Kasbi et al. (2022)	Oman	188	K, CMA, TGP	GDD/ID	27%	[67]
Levchenko et al. (2022)	Russia	133	K or CMA	Non-specific GDD/ID	27%	[8]
Chen et al. (2021)	Taiwan	49	CMA	Unexplained moderate–severe ID	51%	[47]
Nouri et al. (2021)	Iran	61	K	Unexplained ID/DD	66%	[68]
Valentino et al. (2021)	Italy	84	CMA	ID, without ASD	39%	[69]
Hu et al. (2019)	Iran	404	NA	Unexplained ID in consanguineous family	54%	[70]
Xiao et al. (2018)	China	33	CMA	Unexplained ID/DD	57%	[71]
Studies with trio or familial approach						
Guo et al. (2021)	China	21	NA	ID	42%	[72]
Hiraide et al. (2021)	Japan	101	NA	Unexplained ID/DD	54%	[73]
McSherry et al. (2021)	Turkey	21	NA	Clinical suspicion of non-syndromic ARID	48%	[74]
Taskiran et al. (2021)	Turkey	59	CMA	ID, born to consanguineous parents	49%	[75]
Xiang et al. (2021)	China	17	NA	Unexplained ID	59%	[76]
Harripaul et al. (2018)	Pakistan and Iran	192	CMA	Unexplained ID in consanguineous family	46%	[77]
Snoeijen-Schouwenaars et al. (2018)	Netherland	100	Single-gene testing	Unexplained Epilepsy and ID	25%	[78]
Zhao et al. (2018)	Sweden	28	NA	ID/DD with dysmorphic features/congenital anomalies	21%	[79]

Abbreviations: AR = autosomal recessive; CMA = chromosome microarray; DD = developmental delay; DY = diagnostic yield; GDD = global developmental delay; ID = intellectual disability; K = karyotyping; N = number of cases with intellectual disability/global development delay/developmental delay; NA = not available; Ref. = reference; TGP = targeted gene panel.

**Table 6 children-10-00501-t006:** Genome sequencing of individuals with intellectual disabilities/developmental disabilities in recent years.

**Study**	**Country**	**N**	**Previous** **Investigation**	**Cohort** **Phenotype**	**DY**	**Ref.**
Abe-Hatano et al. (2022)	Japan	45	NA	ID	24%	[88]
Zahir et al. (2021)	Canada	8	CMA	Moderate–severe ID with brain malformation	63%	[89]
Sun et al. (2017)	China	100	CMA, ES	GDD/ID	21%	[7]

Abbreviations: CMA = chromosome microarray; DY = diagnostic yield; ES = exome sequencing; GDD = global developmental delay; ID = intellectual disability; N = number of cases with intellectual disability/global development delay/developmental delay; NA = not available; Ref. = reference.

## Data Availability

Not applicable.

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
