# Peer review of "Genome-Wide Sequencing Modalities for Children with Unexplained Global Developmental Delay and Intellectual Disabilities—A Narrative Review"

_children, 2023, doi:10.3390/children10030501_

Round 1

Reviewer 1 Report

This review summarizes recent progress on the etiologic diagnoses of GDD/ID. The review is well written and language is excellent. Some minor edits are needed to make the review more readable to readers who are not in the close field. 

1. In the intro, the authors should consider doing a more thorough introduction of the definition of GDD and ID, e.g., what are the developmental domains that the authors refer to? How is unexplained GDD/ID defined and why is it important to understand the etiology.

2. In the first paragraph of 3.2.2, the authors mentioned variant calling pipelines for ES without explaining exactly how they are superior to GS. The readers may be confused.

3. Line 257-258, reference is needed.

Author Response

This review summarizes recent progress on the etiologic diagnoses of GDD/ID. The review is well written and language is excellent. Some minor edits are needed to make the review more readable to readers who are not in the close field. 

In the intro, the authors should consider doing a more thorough introduction of the definition of GDD and ID, e.g., what are the developmental domains that the authors refer to? How is unexplained GDD/ID defined and why is it important to understand the etiology.

A1-1

Thank you for your kind advice. We have included descriptions on developmental domains and unexplained GDD/ID as you have kindly suggested.

Line 33-35

Childhood neurodevelopment is complex and is interwoven with genetic, biologic, and environmental factors. A simplified schema of developmental domains constitutes of cognitive, language, socio-emotional, and motor neurodevelopment.

Line 43-45

GDD/ID can be secondary to known genetic or chromosomal disorders, or be secondary to known antenatal or perinatal insults of the brain. However, the majority of GDD/ID remains idiopathic, or unexplained.

In the first paragraph of 3.2.2, the authors mentioned variant calling pipelines for ES without explaining exactly how they are superior to GS. The readers may be confused.

A1-2

Thank you for raising the concern. The widespread clinical application of ES in recent decades has accelerated the expansion of the reference datasets. In the meanwhile, the bioinformatics of GS is just starting to catch up. We meant to say ES therefore has better diagnostic performance in terms of data management because the analytical algorithms are constantly being updated and refinned throughout the years. We have modified the original passage to avoid this misunderstanding.

Line 211-17

The widespread use of ES recently has allowed rapid expansion of the sequencing library, and refinement of existing analysis protocols. The analytical steps including the primary filtering variant calling pipeline, and the secondary filtering in silico predictive programs have been augmented to interpretate the coding regions and their vicinity efficiently. Because GS is just starting to gain popularity, the bioinformatics and the associated data managements of GS are relatively immature compared to those of ES.

Line 257-258, reference is needed.

A1-3

We apologize for this oversight in the original manuscript. We have updated the reference. Thank you again for pointing this out.

Line 264-266

Overall, compared to ES, the extrapolation of the additional information by GS is estimated to enhance the diagnostic rate of ES by 9% to 15% [85,86].

Reviewer 2 Report

The authors reviewed recent studies that utelized genome-wide analysis technologies for diagnosing unexplained global developmental delay (GDD) and intellectual disabilities (ID). This is an important and very thorough study, clearly written and with an important conclusion; an evaluation algorithm with exome sequencing (ES) as the first-tier evaluation for unexplained GDD/ID.

Author Response

Thank you for the kind comments from reviewer #2. Your kind words are very much appreciated. 

Reviewer 3 Report

The manuscript of Hsin-Ju Ko and Chen is a well-written narrative review on the diagnostic yield of using different laboratory methods to identify the genetic background of global developmental delay (GDD) and intellectual disabilities (ID). The authors analyse also the cost-effectiveness of the methods and propose a diagnostic algorithm. 

In the introduction, a sufficient background of the disease etiology and spectrum of laboratory methods is provided.  I find it especially useful that the authors describe and present as a Table a key terminology in molecular diagnostics, as especially for clinicians this is important to avoid misunderstandings in this area. In the results section, the authors compare studies and the diagnostic yield obtained with a specific method. In the discussion authors comments on the following aspects: cost-effectiveness, limitations of the current studies, variant interpretation, quality and equity in resource allocation. Finally, a diagnostic algorithm is suggested. The authors summarize their manuscript with concise conclusions.

Comments:

The authors decided to write the manuscript as a narrative review. This brings however a risk of including studies in a biased way. Also, the study design leads to a bias. That is why I would avoid providing a diagnostic yield in % in the abstract.

I would consider presenting cost-effectiveness results in a table and maybe, including them in the results section.

Please include also a comment on the heterogenicity of cohorts there, where the design biased is described (e.g. different results in the tertiary center than in the local centers)

Minor comments:

Table 1 and lines 92 and 94 - please write genes in italics

Line 301-302 to which study does this sentence relates? "In comparison to 301 singletons ES, trio ES had better, though statistically insignificant, diagnostic yield"

Author Response

Point 1: The authors decided to write the manuscript as a narrative review. This brings however a risk of including studies in a biased way. Also, the study design leads to a bias. That is why I would avoid providing a diagnostic yield in % in the abstract.

Response 1: We absolutely agree with the reviewer’s comment regarding the potential bias of narrative reviews. It is the inevitable flaw of narrative review. But narrative review has the advantages of providing a more extensive discussion regarding the topic of interest. While meta-analysis studies and systemic reviews are more objective, only limited numbers of studies fulfilling the stringent selection criteria may be reviewed. However, when evaluating novel technologies that are under development such as genome sequencing and analysis, majority of the available studies often has only a small number of participants. In addition, the methodologies often varied widely between studies and labs, making direct comparison difficult. Therefore, these original studies are often excluded from meta-analysis. Narrative review allows non-discriminatory inclusion of all the original articles available. In this respect, narrative review may offer a more wholistic picture, and may be more suitable for new technologies when information are still scarce.

Nevertheless, we do understand the limitation of narrative reviews and do agree with you concern. We would love to follow up this narrative review with a meta-analysis in the future. We are especially interested in the diagnostic yields of the different genetic analysis methodologies when more studies on GS are available.

Because of the unavoidable bias of narrative review, we do agree with you heartly the existing studies may insufficiently presented in our article. We therefore removed the description on diagnostic yield from the abstract as you have suggested.

Point 2: I would consider presenting cost-effectiveness results in a table and maybe, including them in the results section.

Response 2: Thank you for the kind suggestion. A table listing the cost-effectiveness of the different methodologies would definitely ease the decision making process for ordering providers and family. We agree with the reviewer full heartily.

However, to compose such table, it would require information on the costs of genetic studies. There are limited literatures regarding this information. Unfortunately, we are only familiar with the costs of genetic analysis in the East-Asia. A table on the local costs of genetic evaluations would not be able to reflect the cost-effectiveness in other regions including North America, Europe, and Australasia. In addition, with recent advancements and widespread clinical application of genome sequencing, the market is becoming very competitive and we are seeing a significant price drop for the consumers. The price range of genetic studies has become wider at the moment.

Moreover, a comprehensive comparison of cost-effectiveness should ideally be based on consistent diagnostic performances. The diagnostic yields still vary widely between studies, and between different modalities. A direct comparison of the cost-effectiveness between studies will require systemic evaluations.

Nonetheless, as mentioned in the main text (line319-321), we believe in the very near future both the costs and the diagnostic efficacy will start to plateau. More studies regarding this issue will also be available. A thorough comparison of the cost-effectiveness using a more systemic approach will be feasible then. At the moment, regrettably we could only deduce the cost-effectiveness from the limited studies available, and could not elaborate further on this topic.

Point 3: Please include also a comment on the heterogenicity of cohorts there, where the design biased is described (e.g. different results in the tertiary center than in the local centers

Response 3: Thank you for the kind suggestion. As we have pointed out in the manuscript, large variations on the diagnostic yields were observed between different modalities and different studies. Table 3 to 6 listed recent studies on CMA, gene panel, ES, and GS, and included some information on the study design.

When we talked about heterogenicity of the cohorts, we meant the cohort phenotypes and inclusion criteria varied widely. There were studies which performed genetic studies on all referred cases, while some studies enrolled only cases previously screened by other genetic or chromosomal testing. Some studies focused on GDD/ID in consanguineous families, and some on unrelated individuals.

We have added the following to the manuscript in the discussion section.

Line 302-309 

The large discrepancies in diagnostic yield could potentially be attributed by the heterogenous study design and participant inclusion criteria. Table 3 to 6 listed the inclusion criteria of studies reviewed. The clinical phenotypes of the participants varied. For example, some studies recruited subjects with any NDD versus subjects with GDD, and some included participants with or without ASD etc. The participants also had different prior genetic evaluation results. Some studies only included subjects with negative genetic testing, while some tested all participants irrespective of previous test results. Meta-analysis should be performed to provide an objective comparison on diagnostic performance.

Minor comments:

Point 5: Table 1 and lines 92 and 94 - please write genes in italics

Response 4: Thank you for the kind reminder. We have corrected Table 1 and the original manuscript as you have kindly advised.

Line 97-98

ID genes with female-biased burdens and extreme skewing that have been identified to date include DDX3X, MECP2, WDR45, SMC1A, HDAC8 and NHS[13,17].

Point 5 Line 301-302 to which study does this sentence relates? "In comparison to 301 singletons ES, trio ES had better, though statistically insignificant, diagnostic yield"

Response 5: We apologize for the omission in the original manuscript. We have updated the reference in the main text. We were referring to the results of Tan et al. "A head-to-head evaluation of the diagnostic efficacy and costs of trio versus singleton exome sequencing analysis." European Journal of Human Genetics 27(12): 1791-1799. In Tan’s cohort, the The diagnostic rate of singleton ES was 11/30 (36.7%) compared to 12/30 (40.0%) by trio ES.

Line 310:

In comparison to singleton ES, trio ES had better, though statistically insignificant, diagnostic yield [64].
